# Integrated Transcriptional and Metabolomic Analysis of Factors Influencing Root Tuber Enlargement during Early Sweet Potato Development

**DOI:** 10.3390/genes15101319

**Published:** 2024-10-14

**Authors:** Yaqin Wu, Xiaojie Jin, Lianjun Wang, Jian Lei, Shasha Chai, Chong Wang, Wenying Zhang, Xinsun Yang

**Affiliations:** 1Institute of Food Crops, Hubei Academy of Agricultural Sciences, Wuhan 430064, China; 17300720603@163.com (Y.W.); xiaojiejin@hbaas.com (X.J.); wanglianjun@hbaas.com (L.W.); leijian2006@hbaas.com (J.L.); chaishasha2008@hbaas.com (S.C.); 2College of Agriculture, Yangtze University, Jingzhou 434025, China; 3Crop Institute of Jiangxi Academy of Agricultural Sciences, Jiangxi Academy of Agricultural Sciences, Nanchang 330200, China; wangchong19409@163.com

**Keywords:** sweet potato, root development, transcriptomics, metabolomics, weighted gene co-expression network analysis (WGCNA)

## Abstract

Background: Sweet potato (*Ipomoea batatas* (L.) Lam.) is widely cultivated as an important food crop. However, the molecular regulatory mechanisms affecting root tuber development are not well understood. Methods: The aim of this study was to systematically reveal the regulatory network of sweet potato root enlargement through transcriptomic and metabolomic analysis in different early stages of sweet potato root development, combined with phenotypic and anatomical observations. Results: Using RNA-seq, we found that the differential genes of the S1 vs. S2, S3 vs. S4, and S4 vs. S5 comparison groups were enriched in the phenylpropane biosynthesis pathway during five developmental stages and identified 67 differentially expressed transcription factors, including AP2, NAC, bHLH, MYB, and C2H2 families. Based on the metabolome, K-means cluster analysis showed that lipids, organic acids, organic oxides, and other substances accumulated differentially in different growth stages. Transcriptome, metabolome, and prophetypic data indicate that the S3-S4 stage is the key stage of root development of sweet potato. Weighted gene co-expression network analysis (WGCNA) showed that transcriptome differential genes were mainly enriched in fructose and mannose metabolism, pentose phosphate, selenium compound metabolism, glycolysis/gluconogenesis, carbon metabolism, and other pathways. The metabolites of different metabolites are mainly concentrated in amino sugar and nucleotide sugar metabolism, flavonoid biosynthesis, alkaloid biosynthesis, pantothenic acid, and coenzyme A biosynthesis. Based on WGCNA analysis of gene-metabolite correlation, 44 differential genes and 31 differential metabolites with high correlation were identified. Conclusions: This study revealed key gene and metabolite changes in early development of sweet potato root tuber and pointed out potential regulatory networks, providing new insights into sweet potato root tuber development and valuable reference for future genetic improvement.

## 1. Introduction

Sweet potato (*Ipomoea batatas* (L.) Lam.) stands as the world’s seventh most significant food crop and is among the crucial root crops globally [1]. It is characterized by high and stable yields, strong adaptability, and abundant nutrition [2]. It is not only an important guarantee for food security due to its drought and barren tolerance [3,4]. Moreover, it has shown great economic value and application potential in food processing [5], feed production [6]**,** and even bioenergy development [7]. As the main economic organ of sweet potato, the expansion of root tubers directly influences the final yield and quality. Therefore, in-depth exploration of the molecular regulatory mechanism of root tuber development is of great significance for enhancing the yield and nutritional value of sweet potato.

The sweet potato vine is its main asexual reproductive organ. The development of tuber roots begins with the adventitious roots of stem nodes, which expand and form starch-containing storage roots [8]. At present, the root development process of sweet potato is divided into three stages: whisker root, firewood root, and tuber root **[9]**, which involves the rapid expansion of root cells and the accumulation of a large amount of carbohydrates. For a long time, scientists have extensively studied the root development of sweet potato tuber from the perspectives of genetics, physiology, and molecular biology. Genetic studies showed that tuber swelling and other traits were regulated by multiple genes. In 2005, Tanaka et al. [10] discovered 10 genes that were highly expressed during the root development of sweet potato tubers, named *SRF1*-*SRF10* (Storage root formation 1–10). It is speculated that they are related to root formation and expansion. In 2008, Tanaka et al. [11] identified three *KNOXI* genes for the first time and named them *Ibkn1*, *Ibkn2,* and *Ibkn3*, respectively. Tissue expression analysis indicated that all three genes were highly expressed in the early stage of root tuber expansion. Physiological studies revealed the mechanism of carbohydrate accumulation and hormone balance in the process of root expansion. Wang et al. [12] determined that carbohydrate accumulation was highly related to root size by measuring the contents of starch and sucrose in different stages of sweet potato. In recent years, along with the swift progress of high-throughput sequencing technology, transcriptomes and metabolomics have gradually turned into effective tools for analyzing complex biological processes of crops. This provides a new perspective for uncovering the molecular basis of physiological processes such as crop growth and development as well as stress response.

Transcriptome studies have successfully identified a number of candidate genes related to root tuber formation and development. Dong et al. [13] discovered that genes related to starch biosynthesis were highly up-regulated in storage roots, additionally, genes associated with plant hormone biosynthesis were identified. It was observed that most of the genes related to IAA, CTK, and ABA biosynthesis in storage roots were up-regulated. Song et al. [14] found that starch branching enzyme I may play a key role in storage root swelling by regulating starch formation and content. Firon et al. [15] found that the expression of starch biosynthesis genes rose in the early stage of root tuber formation. Regarding metabonomics, some studies have indicated that shading cultivation can boost the pigment accumulation of purple-fleshed sweet potato storage roots by more than 20%. Moreover, three key enzymes, namely *CHS*, *ANS*, and *3GT*, have been found in the storage roots. These enzymes affect the anthocyanin content in the roots by influencing the metabolic pathway of flavonoids [16]. Meanwhile, reports on the coloring mechanism of anthocyanin in purple sweet potato [17] and the optimization of sweet potato varieties through comparing the nutritional components of different sweet potato varieties have been presented [18]. In spite of this, most of these studies focused on the functional verification of specific genes or specific metabolites, and metabonomics were less involved in regulating the growth and development of sweet potato root tuber. The systematic analysis of the dynamic changes of transcriptome and metabolite in different developmental stages of sweet potato root is still insufficient.

In view of this, this study systematically analyzed the transcriptome and metabolome changes of the early sweet potato root tuber ‘G8017’ in different stages of early development by combining phenotypic and anatomical observations and explored the interaction network between transcription and metabolism, offering new perspectives on the mechanism of sweet potato root tuber expansion. It also established a solid scientific foundation for enhancing the yield and quality of sweet potato through molecular design breeding in the future.

## 2. Materials and Methods

### 2.1. Experimental Materials and Treatment

The tested material is sweet potato variety ‘G8017’, which is an early maturing variety selected in group hybrid seed production. In June 2023, the sweet potato pot farm of the Grain Crop Research Institute of Hubei Academy of Agricultural Sciences planted ‘G8017’ in flowerpots with caliber 28 cm and high 36 cm. Samples were collected on the 7th day (S1), 14th day (S2), 21st day (S3), 28th day (S4), and 35th day (S5) after cutting. Five plants were randomly selected at each development stage. The roots and aboveground plants were sampled. After root washing, the thicker adventitious roots were selected from each plant, and the thickest part in the middle was cut off about 1 cm in length, which was stored in FAA fixed solution for paraffin section production. Some of them were dried at 80 °C for 2 days to obtain stable dry weight for the analysis of starch and sugar content, while the rest were immediately frozen in liquid nitrogen and stored in a refrigerator of −80 °C for transcriptome sequencing and metabonomic analysis. Three biological replicates were taken in each growth period.

### 2.2. Phenotypic and Anatomical Observation

The physiological indexes such as stem length, stem diameter, stem weight, leaf weight, root number, main root diameter, and root fresh weight were measured after root and aboveground plants were sampled. The root differentiation process of sweet potato tubers was observed by the paraffin section method, and the development of primary cambium and the activity of parenchyma cells were observed by saffron and fast green staining sections. The cross-sectional anatomical structure of the root system was observed and photographed by a NIKON MODEL E-CLIPSE Ci- L-type optical microscope. The cross-sectional diameter, cross-sectional area, cortical thickness, primary xylem number, and secondary xylem number were measured by Image-ProPlus 6.0 image processing software and manual counting.

### 2.3. Determination of Sucrose, Total Soluble Sugar and Starch

The roots of sweet potato in different growth periods were washed and dried. The contents of sucrose, soluble sugar, and starch were determined by anthrone colorimetry. Accurately weigh 0.1 g of the dried sample in a 10 mL centrifuge tube, add 80% ethanol to 5 mL, heat and extract in a water bath at 80 °C for 20 min, remove and cool, centrifuged at 4000× *g* rpm for 5–10 min, transfer the supernatant to the 50 mL capacity bottle, extract twice in the same method, merge the supernatant in the 50 mL capacity bottle, fix the volume to the scale, and shake well (this is liquid A for the determination of sucrose and soluble total sugar). Add 4 mL of water to the precipitation, add 2 mL of 9.2 N perchloric acid, and extract in a boiling water bath for 20 min; remove and cool. Centrifuge for 5–10 min. The supernatant is transferred to a 50 mL volumetric flask. Then add 5 mL of water to the precipitation, add 1 mL of 9.2 N perchloric acid, extract in a boiling water bath for 20 min; remove and cool, centrifuge for 5–10 min, and the supernatant is transferred to a 50 mL volumetric flask. Then wash and precipitate with water for 1–2 times, all transferred to the volumetric bottle, fixed volume to the scale, shake well (this is liquid B for starch determination). Sucrose determination: accurately absorb 100 μL of liquid A in the test tube, add 30% KOH of 0.1 mL, put it in boiling water bath for 10 min, cool to room temperature, add 3 mL anthrone reagent, 10 min in boiling water bath, 620 nm colorimetry after cooling. Determination of total soluble sugar: 100 μL of solution A was accurately absorbed into the test tube, 3 mL of anthrone reagent was added, boiled in a water bath for 10 min, and the 620 nm colorimetric method was used after cooling. Starch determination: accurately absorb 100 μL of liquid B in the test tube, add 3 mL anthrone reagent, 10 min in boiling water bath, cool, and 620 nm colorimetric.

### 2.4. Transcriptome Analysis

#### 2.4.1. RNA Extraction and Identification

Total plant RNA was extracted with the RNA Prep Pure Plant Kit (Tiangen, Beijing, China). RNA concentration and purity was measured using NanoDrop 2000 (Thermo Fisher Scientific, Wilmington, DE). RNA integrity was assessed using the RNA Nano 6000 Assay Kit of the Agilent Bioanalyzer 2100 system (Agilent Technologies, CA, USA).

#### 2.4.2. Library Construction and Sequencing

The total amount of database built at the beginning of each sample is 1 μg. Employ magnetic beads with oligo (DT) to capture mRNA from the total RNA. The first-strand cDNA is synthesized first, followed by the synthesis of the second-strand cDNA. The protruding end was repaired and transformed into a flat end by exonuclease and polymerase activities. After the adenosine at the 3′ end of the DNA fragment was acidified, the NEBNext joint with a hairpin ring structure was connected. AMPure XP system (Beckman Coulter, Beverly, USA) were used to purify the library fragments. Then, add 3 μL USER Enzyme (NEB, USA) and incubate at 37 °C for 15 min and react at 95 °C for 5 min before PCR. Subsequently, PCR was performed with high-fidelity DNA polymerase, universal PCR primers, and index (X) primers. At last, PCR products were purified (AMPure XP system) and library quality was assessed on the Agilent Bioanalyzer 2100 system. The library was sequenced on the Illumina NovaSeq platform to generate 150 bp double-terminal sequences.

#### 2.4.3. Data Quality Control, Sequence Alignment and Comments

The original reads were further processed using the bioinformatics analysis platform BMKCloud (www.biocloud.net). The raw data in Fastq format is initially processed by internal Perl scripts. In this step, valid data (clean data) is obtained by removing a sequence containing connectors, a sequence containing poly-N, and a low-quality sequence from the original data. Simultaneously, Q20, Q30, GC content, and sequence repeat level were calculated. All downstream analyses are based on high quality clean data. The effective data were compared to the reference genome sequence by Hisat2 tool software. According to the reference genome, only a perfectly matched or a mismatched sequence is further analyzed and annotated. The method of assembling StringTie transcripts based on reference annotations (RABT) identifies known transcripts and predicts new transcripts from Hisat2 alignment results. Gene function is annotated through sequence alignment with reference to the following databases: Nr; Pfam; KOG/COG; Swiss-Prot; KO; and GO.

#### 2.4.4. Transcriptome Data Analysis

The quantification of gene expression level is assessed by the number of sequences mapped per thousand base transcripts per million. The formula is as follows: {cDNA Fragments\over {Mapped Fragments (Millions) * TranscriptLength (kb)}}. DESeq2 is harnessed to analyze the differential expression within the comparison group. DESeq2 makes use of a model founded on negative binomial distribution to determine the differential expression in gene expression data. The corrected p value is derived by applying Benjamini and Hochberg to control the false discovery rate. Genes with a p value less than 0.01 and a fold change of at least 2, as adjusted by DESeq2 analysis, are designated as differentially expressed. The GO enrichment analysis of differentially expressed genes (DEGs) is realized by the clusterProfiler package based on the non-central hypergeometric distribution of Wallenius, which can rectify the deviation of gene length in differential genes. The KOBAS [19] database and clusterProfiler software (Version 4.4.4) are utilized to analyze the enrichment of DEGs in the KEGG pathway.

### 2.5. Metabonomic Analysis

#### 2.5.1. Extraction of Samples

Weigh the 50 mg sample, add 1000 μL of extract containing an internal standard (with a methanol-acetonitrile-water volume ratio of 2:2:1 and an internal standard concentration of 20 mg/L), Perform vortex mixing for 30 s and add steel ball, 45 Hz grinder to treat 10 min, ultrasonic 10 min (ice water bath), stand at minus 20 °C for one hour, centrifuged at 12,000× *g* rpm at 4° C for 15 min, remove 500 μL of supernatant in an EP tube, and dry the extract in a vacuum concentrator. Add 160 μL extract (acetonitrile volume ratio: 1:1) to the dried metabolite, vortex for 30 s, ultrasound in an ice water bath for 10 min, centrifuged at 12,000× *g* rpm at 4 °C for 15 min, remove 120 μL of supernatant into a 2 mL injection bottle, and take 10 μL of each sample and mix it into the QC sample and test it on the machine.

#### 2.5.2. Conditions for Metabolite Extraction and Detection in Non-Targeted Metabolic Group

Metabonomic analysis was conducted using LC/MS. The system comprised Waters Aquity I PLUS ultra-high-performance liquid phase in tandem with Waters Xevo G2-XS QTof high resolution mass spectrometer. Analysis conditions: In positive ion mode, mobile phase A is 0.1% formic acid aqueous solution, and mobile phase B is 0.1% formic acid acetonitrile solution. In negative ion mode, mobile phase A is also 0.1% formic acid aqueous solution, and mobile phase B is 0.1% formic acid acetonitrile solution. The injection volume is 1 μL. The high-resolution mass spectrometer can collect first- and second-stage mass spectra data in the MSe mode controlled by the acquisition software (MassLynx V4.2). In each data acquisition cycle, dual-channel data acquisition for low collision energy and high collision energy can be performed simultaneously. The range of low collision energy is 2 V. The range of high collision energy is from 10 to 40 V. The scanning frequency is 0.2 s. The parameters of the ESI ion source are as follows: capillary voltage: 2500 V (in positive ion mode) or 2000 V (in negative ion mode); cone hole voltage: 30 V; ion source temperature: 100 °C; desolvent gas temperature: 500 °C; reverse blowing flow rate: 50 L per hour; desolvent gas flow rate: 800 L per hour; the collection range of mass-to-charge ratio (*m*/*z*) is 50–1200.

#### 2.5.3. Metabolome Data Analysis

After normalizing the information of the original peak area and the total peak area, subsequent analysis was carried out. Principal component analysis and Spearman correlation analysis were employed to judge the repeatability of samples within the group and quality control samples. The classification and pathway information of the identified compounds were searched in KEGG, HMDB, and Lipid Maps databases. According to the grouping information, the difference multiple of each component was computed and compared. The significant difference value of each component was calculated through a *t*-test. The R language package ROPL is utilized for OPLS-DA modeling. Two hundred permutation tests are conducted to verify the reliability of the model. The VIP value of the model is calculated by multiple cross-validations. The differential metabolites of the OPLS-DA model were screened by combining the difference multiple, *p* value, and VIP value. The screening criteria were FC > 1, *p* < 0.05, and VIP > 1. To analyze the changing trend of metabolites, DAM was standardized (z-score) and clustered using K-means. The hypergeometric distribution test was employed to calculate the enrichment of significant differential metabolites in the KEGG pathway.

### 2.6. Combined Transcriptome and Metabolome Analysis

The transcriptome and metabolomic data using the lotaustralin cloud platform “https://international.biocloud.net”(accessed on 13 October 2024) by the weighted gene express network (WGCNA) is analyzed. Quantitative values of genes and metabolites were analyzed by correlation analysis. The genes and metabolites with pmur values of 0.05 and correlation threshold −>0.8 were screened using the Pearson correlation calculation method on the McWei cloud platform “https://cloud.metware.cn”(accessed on 13 October 2024), and the correlation coefficient clustering heat map was drawn.

## 3. Results

### 3.1. Phenotypic Identification at Different Growth Stages

The root length, taproot diameter, root number, stem length, stem diameter, stem weight, leaf weight, aboveground weight, and root fresh weight of ‘G8017’ at five developmental stages were investigated (Table 1, Figure 1). The root length and taproot diameter in the table were the average values of the measured data of the five differentiated roots of each plant. In order to have a more intuitive understanding of the changes of different indicators in different development stages, a bar chart was drawn based on the average value of the information in the table. It can be seen from the chart that taproot diameter, stem length, stem diameter, leaf weight, stem weight, and root fresh weight all showed an upward trend with the development of sweet potato seedlings. Among them, the most significant change is from the second period to the fourth period, with a significant increase in all indicators.

### 3.2. Anatomical Analysis of Differentiated Roots of Sweet Potato

Figure 2 shows the morphology of the differentiated roots of sweet potato ‘G8017’ on days 7, 14, 21, 28, and 35 of growth, as well as the cross-sectional diagrams of the first four stages under a 4 × 10-fold microscope (one-quarter cross-section was taken at the fourth stage). Table 2 shows the internal structural quantitative indexes of root differentiation at the 7th, 14th and 21st days of sweet potato growth. On day 7 of growth, the primary xylem was the main conducting tissue, responsible for transporting water and inorganic salts from the soil to other parts of the plant. At this stage, the protoxylem cells are more active and arranged in six radial bundles along the center. With the growth of sweet potato (on the 14th day of growth), the number of cells in the primary xylem increased, and the transport ability was enhanced. The primary cambium began to differentiate into secondary xylem, and these cells gradually formed new transport tissues outside the primary xylem. Thin cortical tissue gradually increased in the primary cambium region, and these cells had a high ability to divide, providing a new source of cells for the growth of sweet potato. On day 21 of growth, the number of primary xylem and secondary xylem cells increased further, and the transport ability was significantly enhanced. The formation of secondary xylem makes the roots of sweet potato stronger. The cells of the primary phloem also continue to grow and divide to maintain the normal transport of photosynthetic products. Thin cortical tissue continued to increase in the primary cambium region and began to differentiate into new cells in the secondary cambium region. On the 28th day of growth, the number of cells in the primary xylem and secondary xylem reached a relatively stable level, and the transport capacity reached the maximum. The roots of the sweet potato were already quite robust at this stage. The number of cells in the primary phloem also tended to be stable, maintaining the normal transport of photosynthates.

### 3.3. Changes of Sucrose, Soluble Sugar, and Starch during the Development of Sweet Potato

The content of soluble sugar is an important quality characteristic of sweet potato and one of the main indicators for the breeding of new varieties. It can be seen from Figure 3 that in the early stage of sweet potato root development, the change trend of sucrose and total soluble sugar content is basically the same without significant change. Starch is the main component of sweet potato root and the most important factor affecting the expansion of sweet potato root. It can be seen from the figure that starch content showed a steady upward trend with the growth and development of sweet potato roots, indicating that starch continued to accumulate after the roots began to develop, and the expansion of sweet potato roots was transformed and filled by starch.

### 3.4. Transcriptome Analysis of Sweet Potato Roots at Different Developmental Stages

Eukaryotic reference transcriptome (RNA-seq) analysis was performed on the root samples of sweet potato ‘G8017’ at different growth stages. The results of transcriptome sequencing showed that the GC content of each sample was between 45.79 and 47.01%. After the joint sequence was removed, a total of 92.64 GB of clean data was obtained. The clean data of each sample reached 5.82 GB Q30 base percentage and above, and the sequencing result was good. Comparing the clean reads of different samples with the reference genome (Appendix A), the results show that the comparison rate of each sample is more than 79.24%, and the alignment results are normal, which meets the requirements of further analysis.

With Fold Change ≥ 2 and FDR < 0.01 as screening criteria, the gene expression of the five phases was divided into four comparison groups for differential expression analysis and statistics (Figure 4a). A total of 13,669 DEGs were obtained: the statistical results showed that there were 2411 (882 up-regulated and 1529 down-regulated), 5214 (2770 up-regulated and 2494 down-regulated), 4679 (1892 up-regulated and 2787 down-regulated), and 1365 (310 up-regulated and 1055 down-regulated) DEGs in S1 vs. S2, S2 vs. S3, S3 vs. S4, and S4 vs. S5 groups, respectively. The statistical results showed that the total number of DEGs increased first and then decreased. A Venn diagram of DEGs was drawn for the four comparison groups (Figure 4b). The S2 vs. S3 comparison group had the largest number of DEGs (2762), while the S4 vs. S5 group had the smallest number (504). There were 51 DEGs shared by the four groups (Appendix A). The results showed that a large number of gene transcription and translation may be activated around days 15–25 of root development, while gene transcription and translation may be inhibited after day 30.

In order to further understand the molecular function of DEGs, the functional enrichment results of GO showed that the DEGs in the five periods were annotated into three categories: biological processes, cellular components, and molecular functions, and the four comparison groups were mainly annotated in cellular processes, metabolic processes, biological regulation, cellular anatomical entities, intracellular, binding, and catalytic activity. The results of KEGG enrichment analysis showed that the differential genes of S1 vs. S2, S3 vs. S4, and S4 vs. S5 were all involved in phenylpropane biosynthesis. In addition, starch and sucrose metabolism and glutathione metabolism were enriched in S3 vs. S4, monoterpenoid biosynthesis was enriched in S1 vs. S2, glycolysis/gluconeogenesis and carbon metabolism were enriched in S2 vs. S3, and MAPK signaling pathway was enriched in S4 vs. S5.

### 3.5. Identification of WGCNA Modules of Differentially Expressed Genes

In order to understand the regulation of DEGs during sweet potato development, we used weighted gene co-expression network analysis (WGCNA) to study the DEG co-expression network. The expression threshold was set as 1, the module similarity threshold as 0.01, the minimum number of genes in the module as 30, and the differential genes were divided into eight modules (Figure 5). The S3 period is the key period of root expansion of sweet potato. KEGG enrichment analysis was performed on genes of black module (*r* = 0.99, *p* = 0.002), brown module (*r* = 0.92, *p* = 0.03), and red module (*r* = 0.89, *p* = 0.04) (Appendix A). The results showed that the genes in the black module were significantly enriched in the fructose and mannose metabolism, pentose phosphate pathways, and caffeine metabolism. The genes in the brown module were significantly enriched in the pathways of seleno compound metabolism, valine, leucine, and isoleucine degradation, and the biosynthesis of various secondary metabolites. The genes in the red module were significantly enriched in glycolysis/gluconeogenesis, carbon metabolism, the pentose phosphate pathway, and other pathways (Appendix A).

### 3.6. Transcription Factor Analysis

Statistically and analyzing the differentially expressed transcription factor genes in the transcriptome (Figure 6), a total of 67 transcription factors (TF) were identified, among which the AP2/ERF-ERF (178), NAC (178), bHLH (175), MYB (160), and C2H2 (149) families were significantly enriched. The results showed that these significantly changed TFs may play an important role in the root expansion process of sweet potato.

### 3.7. Accumulation and Difference Analysis of Metabolites during Root Development of Sweet Potato

Differences in the types and contents of metabolites in sweet potato roots may be important factors affecting various traits, including root tuber enlargement. The metabolites in samples were identified by LC-MS platform non-targeted metabolome technology. In different stages of sweet potato root development, a total of 4695 metabolites were detected. The top 20 types with the most annotations were selected. It mainly includes carboxylic acids and derivatives, prenol lipids, fatty acyl, organooxygen compounds, steroids and steroid derivatives, etc. (Table 3).

PCA of metabolites in the comparison group of sweet potato ‘G8017’ at different developmental stages showed that PC1 and PC2 could completely distinguish the four combined treatments (Figure 7a,b), indicating that there were significant differences in the metabolic groups of sweet potato roots at different developmental stages, among which there was a large gap between the S3 group and the S4 group. It was speculated that the metabolites in the root of sweet potato changed significantly during the 20–30 d stage of growth and development. In order to more clearly understand the law of metabolite changes in sweet potato roots at different developmental stages, the differential metabolites were screened and analyzed. The results showed that (Figure 7c), the comparison groups S1 vs. S2, S2 vs. S3, S3 vs. S4, and S4 vs. S5 obtained 952 (556) up-regulated metabolites, respectively. There were 396 down-regulated, 967 (515 up-regulated, 452 down-regulated), 2046 (783 up-regulated, 1263 down-regulated), and 624 (221 up-regulated, 403 down-regulated) differential metabolites. Among them, there were more differential metabolites in S3 vs. S4, which was consistent with the results of PCA, that is, the transition from the S3 stage to the S4 stage was an important turning point in sweet potato root development. The Venn diagram summarized the differences in metabolite changes between the different comparison groups (Figure 7d), with 244, 636, and 360 metabolites jointly differentially expressed in the S1 vs. S2 and S2 vs. S3, S2 vs. S3 and S3 vs. S4, and S3 vs. S4 and S4 vs. S5 comparison groups, respectively. Twenty-nine metabolites were expressed in all five periods of the sample (Appendix A).

### 3.8. K-Means Cluster Analysis of Differential Metabolites

In order to analyze the changing trend of metabolite content during root development of sweet potato, the relative contents of all different metabolites identified in each group were standardized according to the screening criteria, and then K-means cluster analysis was carried out (Figure 8a). Class1 contains 450 dam, which accumulates mainly in the S1–S3 stage of root development and decreases in the S4 and S5 stages (Figure 8b,c). The expression patterns of class4 and class1 were similar, showing a significant downward trend with the development of roots (Figure 8b), which was contrary to the metabolite accumulation pattern of class3. Class3 was significantly up-regulated from the S3 stage and contained a large number of carboxylic acids and derivatives, organooxygen compounds, and prenol lipids (Figure 8b grad). In addition, the metabolites of class2, class5, and class6 were mainly accumulated in S2, S4, and S3, respectively. The results showed that lipids, organic acids, and organic oxides accumulated differently in different growth periods, which may be an important factor affecting the root expansion of sweet potato.

### 3.9. Identification of Differential Metabolite WGCNA Modules

To further understand the relationship between metabolite accumulation and root tuber development, weighted gene co-expression network analysis (WGCNA) was performed. The expression threshold was set to 1, the module similarity threshold to 0.5, and the minimum gene number to 30. The differential metabolites were divided into 12 modules (Figure 9). The metabolome data showed that the differential metabolites accumulated the most during the S3–S4 period, and it was inferred that the dynamic changes of metabolites in sweet potato roots at this stage were significant. KEGG enrichment analysis was performed for green modules (r = 0.99, *p* < 0.001) and turquoise modules (r = 0.99, *p* = 0.001) (Appendix A). In green modules, amino sugar and nucleotide sugar metabolism and flavonoid biosynthesis pathways were significantly enriched. Alkaloid biosynthesis, pantothenic acid and CoA biosynthesis, polyketide sugar unit biosynthesis, and other pathways were enriched in the turquoise module, and the metabolites of significant enrichment pathways were sorted and classified (Appendix A).

### 3.10. Combined Analysis of Transcriptome and Metabolome

Based on the above WGCNA of the transcriptome and metabolome of sweet potato roots at different growth stages, we analyzed the correlation between 55 differential genes and differential metabolites that were screened by the transcriptome and 59 differential metabolites that were possibly related to root expansion. Genes and metabolites with a *p* value < 0.05 and a correlation threshold > 0.8 were screened, and the Pearson correlation calculation method was used to draw a heat map (Figure 10), which included 44 differential genes and 31 differential metabolites. Among them, g797, g792, g30713, g786, g30712, g782, g42961, g42970, g55289, g14527, g14528, g49596, g793, g11788, g778, g11789, and g59572, with at least 25 different metabolites. Those with an extremely significant positive correlation include metabolites dTDP-L-megosamine Vindoline, D-4′-Phosphopantothenate vinorine Allocryptopine, 3-Methyl-2-oxobutanoic Acid, (3R)-3-Hydroxy-2, 3-dihydrotabersonine, 6-Hydroxyprotopine, Thebaine, lsocorypalmine, Morphine, Cathenamine Colcemid, Maltol, 4,21-Dehydrogeissoschizine Methylcanadine, Agroclavine, 6-O-Methyldeacetylisoipecoside, with at least 25 of them there was a significant positive correlation between allogenes.

## 4. Discussion

In the growth process of root crops, the initial stage of root expansion to form storage roots is a crucial period that affects the final yield. To find the critical period of early root development and explore the molecular mechanism that may affect its expansion is the key of current research. Traditional single omics studies often can only provide limited biological information, but the combined analysis of transcriptomics and metabolomics can establish a complete biological pathway from gene expression to metabolites, and deeply understand the molecular mechanism of sweet potato root enlargement.

The changes in phenotypic data, root anatomical characteristics, and physiological indexes observed in the study were closely related to the rapid expansion of roots. In the growth process of the five periods, except for root length and root number, other phenotypic data showed an upward trend, especially in the second to fourth development stages (such as stem length, stem weight, leaf weight, aboveground fresh weight, etc.). Root growth actively promotes the change of aboveground photosynthetic rate. The aboveground material distribution, matter accumulation, and photosynthesis-related indexes also affected the root tuber development to a certain extent [20], indicating that this stage is the most active stage in the early stage of sweet potato root growth. Through dynamic network biomarker (DNB) analysis, it was determined that the early stage of root diameter 3.5 mm was the critical period for sweet potato storage root expansion [21]. In this study, sweet potato root diameter reached 3.46 mm in the S3 stage, which was consistent with previous conclusions. Anatomical observation revealed the dynamic changes of xylem structure: the number of primary xylem increased significantly with the growth of tuber roots, and the formation of secondary xylem provided a structural basis for root tuber growth. The formation of stored roots in plants is closely related to the storage parenchyma of secondary xylem and secondary phloem and the dense growth of primary cortex [22]. In addition, anatomical analysis showed cell differentiation and tissue structure optimization during S3–S4, which further confirmed the effective accumulation and distribution of nutrients in roots during this period. These findings emphasized the importance of root structural changes to the growth of root crops. Starch and soluble sugar content are important factors affecting the quality and yield of sweet potato [23]. Bahaji et al. [24] showed that starch accumulated continuously after root formation and decreased slightly only in the late stage of root expansion. In this study, the starch content maintained an increasing trend throughout the sampling period, while the soluble sugar content, including sucrose, did not change significantly.

In this study, we constructed a global transcription-metabolome dataset of five stages of early sweet potato development and identified 13,669 DEGs and 4695 differentially accumulated metabolites during root development of sweet potato. According to transcriptome data analysis, the two genes (g46667 and g46673) whose expression was up-regulated in the four comparison groups were sporamin (sweet potato root storage protein), which accounted for more than 80% of the total sweet potato root protein [25]. The transcription factors AP2/ERF, NAC, bHLH, MYB, and C2H2 family are significantly enriched, and it has been reported that these transcription factors are involved in regulating the growth and development of plant roots [26,27,28,29,30]. It is speculated that these transcription factors may affect the root expansion of sweet potato by regulating the expression of downstream genes. Multiple groups of differential genes are involved in the phenylpropane biosynthesis pathway, and lignin, one of the secondary metabolites of the phenylpropane biosynthesis pathway, is crucial for the formation and lignification of plant cell walls [31,32]. Some lignin biosynthesis genes are induced after *Arabidopsis* injury, and lignin monomers are deposited in the secondary cell wall to enhance the formation of cell walls [33,34]. *PAL* in phenylpropanoid metabolism is a key factor in inducing and developing sweet potato callus [35], and its activity in sweet potato root development may be closely related to root structural strengthening and morphogenesis. Single terpenoids can affect plant growth regulatory factors, such as the level or activity of plant hormones, and thus affect the expansion of sweet potato roots. GGPP is an important metabolic branch point in the terpenoid metabolic network, regulating the synthesis of primary metabolites such as gibberellin, abscisic acid, and tocopherol, and plays an important role in plant growth and development [36]. The expansion of sweet potato root depends on carbon metabolism. Sugars produced by photosynthesis are transported to the root via phloem, where they are converted into starch accumulation and promote root growth [37]. Fructose-1,6-diphosphatase (FBPase), one of the key enzymes in the gluconeogenesis pathway, is a key enzyme in the Calvin cycle and starch synthesis. The activity of this enzyme can enable yeast cells to switch between glycolysis and the gluconeogenesis pathway [38], and it plays an important regulatory role in photosynthetic carbon assimilation and allocation [39]. Therefore, it is very important for the growth and development of sweet potato. Studies have shown that carbohydrate metabolism and starch biosynthesis function are up-regulated during storage root development of sweet potato [15]. Sucrose synthase is related to tuber/root growth of potato and radish and is a key enzyme in the early development of storage root of radish [40,41]. Glutathione (GSH) is beneficial to root development, and the *RML1* gene encodes the first enzyme of glutathione biosynthesis. GSH has an alleviating effect on *RML1* mutant epigenetic forms, and the G (1) to S phase transition requires sufficient levels of glutathione, and GSH is essential for cell division during postembryonic root development [42]. Previous studies on the regulation of light/sugar induced ion transporter genes have found a sugar sensing pathway dependent on the oxidative pentose phosphate pathway to control root nitrogen and sulfur acquisition through plant carbon status [43]. The above metabolic pathways are enriched in different stages of sweet potato root growth, indicating that these pathways may play a role in different periods to meet the specific needs of root growth and development.

At the same time, through differential metabolite screening and K-means cluster analysis, we also found that some metabolites accumulated differently in different growth periods, such as prenol lipids, carboxylic acids and derivatives, organooxygen compounds, etc., among which isoprenoid substances such as chlorophyll, plant hormones, carotenoids, terpenoids, and coenzyme Q are important substances to maintain plant growth and development [44]. Gibberellin promotes stem and root growth by inhibiting the activity of transcription factors RGA and GAI [45]. Organic acids such as citric acid in sweet potato roots are not only intermediate products of the tricarboxylic acid cycle but also important components of intracellular pH regulation. Indoleacetic acid (IAA) can promote plant root growth and development, and tryptophan is the main precursor of IAA biosynthesis [46]. The transport measurements of plastids isolated from pea roots showed that inorganic phosphate, dihydroxyacetone phosphate (DHAP), 3-phosphoglyceric acid, 2-phosphoglyceric acid, phosphoenolpyruvate, and glucose-6-phosphate (Glc6P) were transported across the membrane in reverse exchange mode. DHAP is an important intermediate in the glycolysis pathway and participates in carbohydrate metabolism [47]. These metabolites may interact with gene expression to regulate the root swelling of sweet potato tubers.

Based on WGCNA analysis, we constructed co-expression networks of genes and metabolites, revealing complex regulatory networks. Different modules represent different biological processes and metabolic pathways, which interact with each other to coordinate the development of sweet potato root. In our study, we found that the S3–S4 period is an important turning point in sweet potato root development, with more differential metabolites, and it echoes the trend of DEGs in transcriptome analysis. This suggests that during this period, changes in gene expression and metabolites coordinate with each other to promote root enlargement. In the WGCNA analysis of DEGs, we divided the DEGs into eight modules and analyzed the black, brown, and red modules that were significantly enriched in the S3 period. The results showed that the main metabolic pathways involved in root development of sweet potato included carbohydrate metabolism (fructose and mannose metabolism, glycolysis/gluconeogenesis, pentose phosphate pathway, carbon metabolism), which provided energy and synthetic raw materials. Amino acid metabolism (degradation of valine, leucine, and isoleucine), nitrogen source, and energy supply; secondary metabolism (biosynthesis of various secondary metabolites) produces plant hormones to regulate growth; and selenium metabolism (metabolism of selenium compounds) enhances antioxidant capacity and protects cells. In the WGCNA analysis of differential metabolites, we divided the differential metabolites into 12 modules and conducted KEGG enrichment analysis on the metabolites of green module and turturine module during the S3–S4 period. The enrichment pathways of amino sugar and nucleotide sugar metabolism, flavonoid biosynthesis, alkaloid biosynthesis, pantothenic acid and CoA biosynthesis, and polyketide sugar unit biosynthesis were found. These pathways are involved in cell structure construction, growth regulation, defense, energy supply, and secondary metabolite synthesis and have synergistic effects in sweet potato root enlargement. We mined 44 differential genes and 31 metabolites based on WGCNA analysis of transcriptome and metabolome and conducted correlation analysis. We found that some DEGs were associated with differentially expressed metabolites, and g797, g792, and other genes were significantly positively correlated with multiple differentially expressed metabolites. dTDP-L-megosamine, Vindoline, and other metabolites were significantly positively correlated with several genes. There could be a variety of mechanisms behind this correlation. On the one hand, DEGs may directly regulate key enzymes in metabolic pathways, thereby affecting the synthesis and accumulation of metabolites. On the other hand, metabolites may also affect gene expression through feedback regulatory mechanisms, and the accumulation of certain metabolites may inhibit or activate the expression of specific genes, thereby regulating the activity of metabolic pathways.

This study not only enhanced our understanding of molecular regulatory networks during root development of sweet potato, but also provided a theoretical basis for improving yield and quality of sweet potato through metabolic engineering and genetic improvement. Future studies can further carry out gene function verification experiments on this basis and clarify the specific functions of key genes and metabolites by means of gene editing, gene cloning, and genetic transformation.

## 5. Conclusions

In this study, the regulation network of root expansion of sweet potato tuber was systematically revealed through transcriptome and metabonomic analysis at different stages of sweet potato root development, combined with phenotypic and anatomical observation. Transcriptome, metabolic group, and early phenotypic data all showed that the S3-S4 stage was the key period for root development of sweet potato. Transcriptome analysis showed that the differential genes of S1 vs. S2, S3 vs. S4 and S4 vs. S5 were enriched in the phenylpropane biosynthesis pathway, and 67 differentially expressed transcription factors, including AP2, NAC, bHLH, MYB, and C2H267, were identified. Metabonomic analysis showed that lipids, organic acids, and organic oxides accumulated differently in different growth periods, which may be an important factor for root expansion of sweet potato tubers. Weighted gene coexpression network analysis (WGCNA) showed that transcriptome differential genes were mainly enriched in fructose and mannose metabolism, pentose phosphate, caffeine metabolism, selenium metabolism, valine leucine and isoleucine degradation, glycolysis/gluconeogenesis, carbon metabolism, and so on. The differential metabolites in the metabolic group are mainly concentrated in amino sugar and nucleotide sugar metabolism, flavonoid biosynthesis, alkaloid biosynthesis, pantothenic acid and coenzyme A biosynthesis, polyketo unit biosynthesis, and so on. Based on WGCNA analysis of gene-metabolite correlation, 44 highly related differential genes and 31 differential metabolites were identified.

## Figures and Tables

**Figure 1 genes-15-01319-f001:**
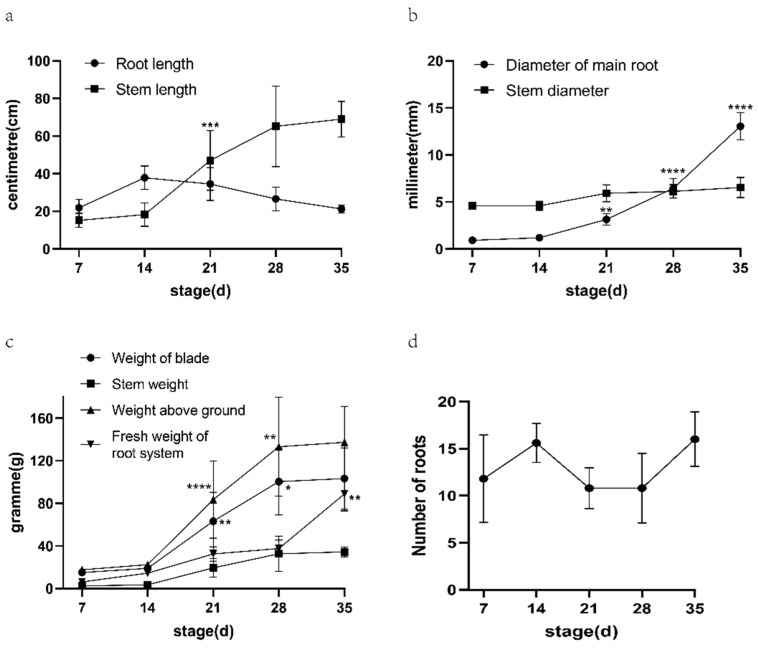
Phenotypic indicators measured at five different developmental stages. (**a**) Root length and stem length, (**b**) Diameter of main root and stem diameter, (**c**) Weight of blade, stem weight, weight above ground, and fresh weight of root system, (**d**) Number of roots. * stands for significance, the more ‘*’ indicates the higher significance.

**Figure 2 genes-15-01319-f002:**
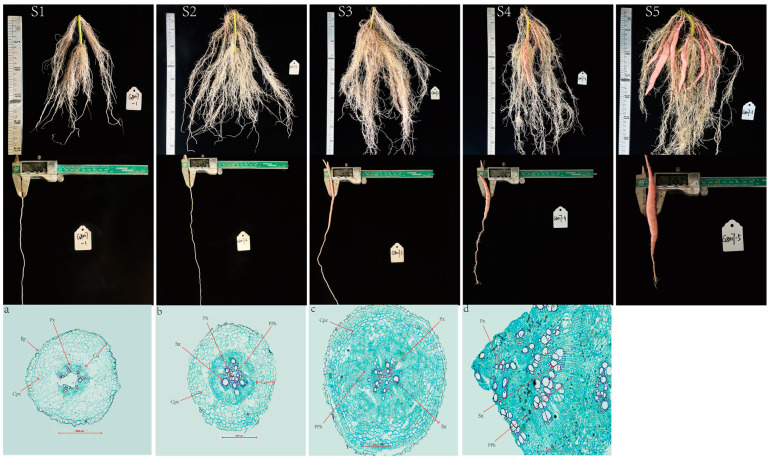
S1–S5 showed the morphologic maps of differentiated roots on days 7, 14, 21, 28, and 35 of growth, respectively. (**a**–**d**) were the cross-sectional structure of differentiated root of sweet potato S1–S4, respectively. Ep: epidermis, Co: cortex, Cpc: cortical parenchyma cells, Px: primary xylem, Sx: secondary xylem, PPh: primary phloem.

**Figure 3 genes-15-01319-f003:**
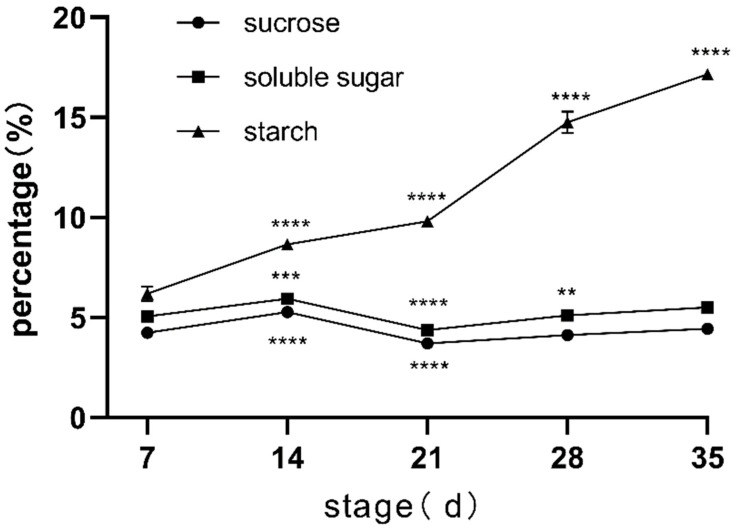
Changes of sucrose, soluble sugar, and starch at different developmental stages. * stands for significance, the more ‘*’ indicates the higher significance.

**Figure 4 genes-15-01319-f004:**
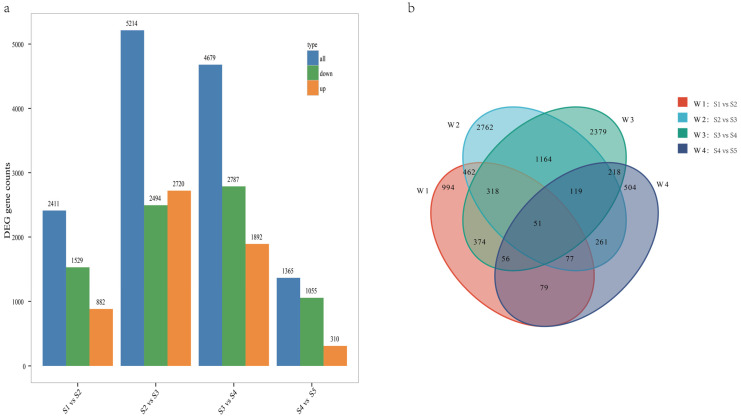
(**a**) The number of differentially expressed genes (DEGs) in comparison groups S1 vs. S2, S2 vs. S3, S3 vs. S4, S4 vs. S5, (**b**) Venn diagram of different developmental stages.

**Figure 5 genes-15-01319-f005:**
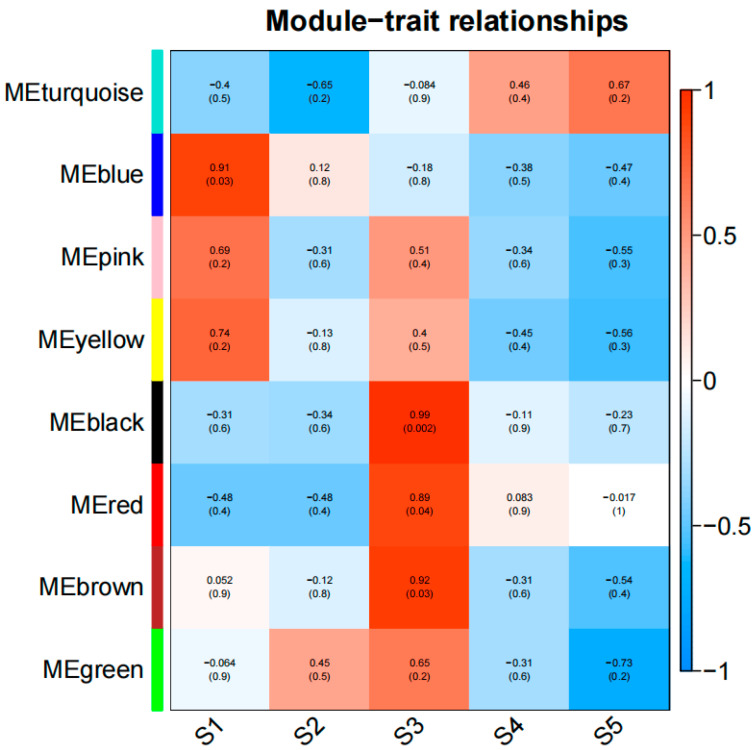
Heat map of correlation between sweet potato root development and differential gene expression patterns at different periods. The heat map shows the expression patterns of eight modules, with color bars representing the expression levels from high (red) to low (blue).

**Figure 6 genes-15-01319-f006:**
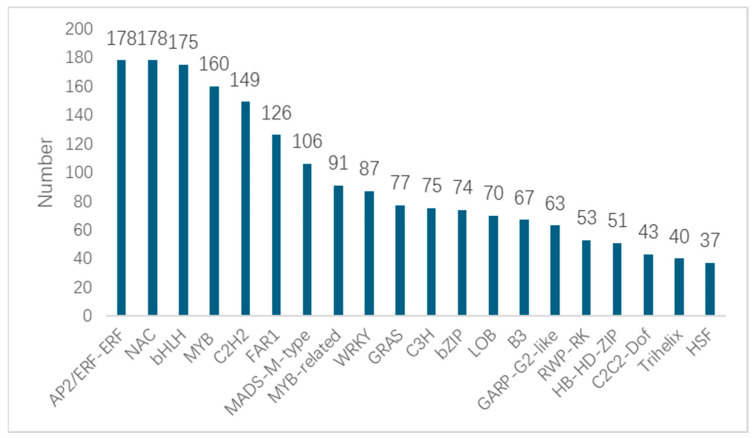
Statistics of types and quantities of transcription factors.

**Figure 7 genes-15-01319-f007:**
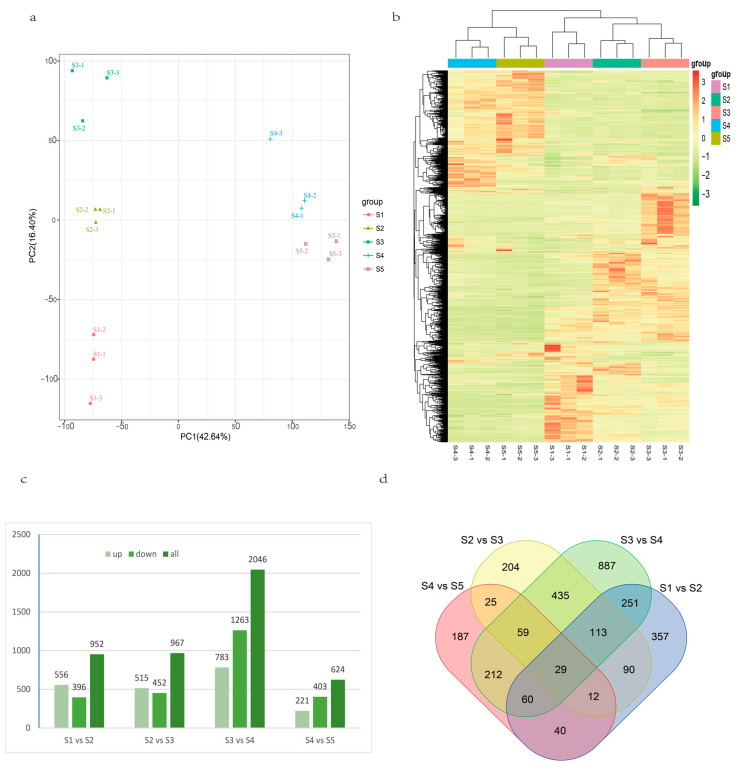
Comparison of metabolites at different developmental stages in sweet potato roots. (**a**) Principal Component Analysis (PCA) score map of all metabolites in the sample, (**b**) differential accumulation of metabolites in the five periods, (**c**) compare the number of DAMs in groups S1 vs. S2, S2 vs. S3, S3 vs. S4, and S4 vs. S5, (**d**) DAMs diagram of different developmental stages.

**Figure 8 genes-15-01319-f008:**
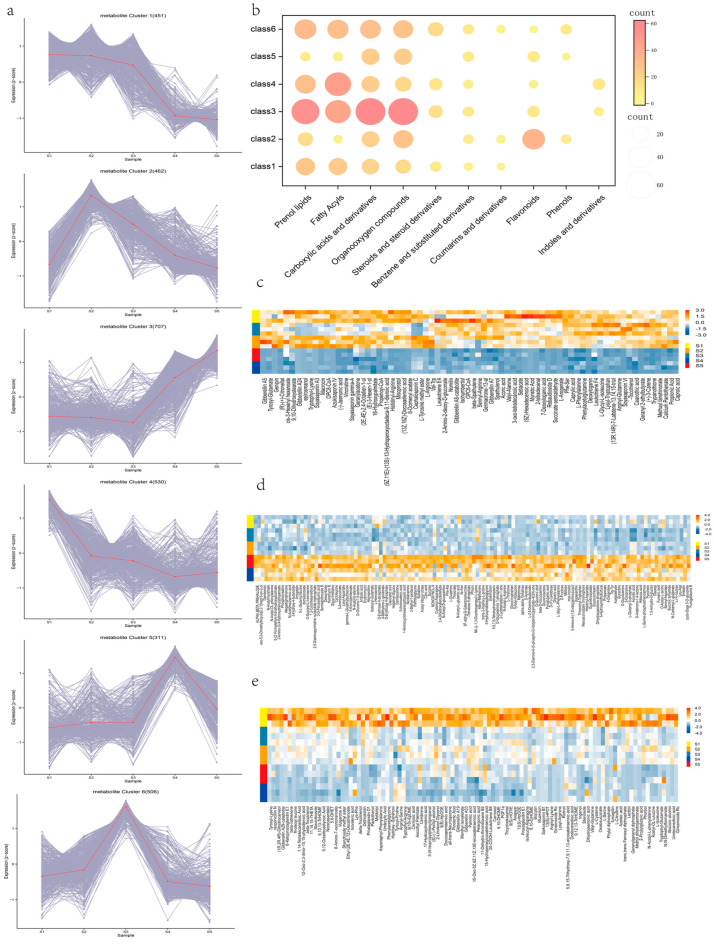
Cluster analysis of differential metabolites. (**a**) K-means cluster analysis of differential metabolites. The gray line represents the intergroup trend of differential metabolite content in each k-means cluster, and the red line represents the average trend, (**b**) The number of metabolites in each category. Nodes represent the number of metabolites from small to large and from shallow to deep, (**c**) class1 metabolite accumulation model, (**d**) class3 metabolite accumulation model, (**e**) class4 metabolite accumulation model.

**Figure 9 genes-15-01319-f009:**
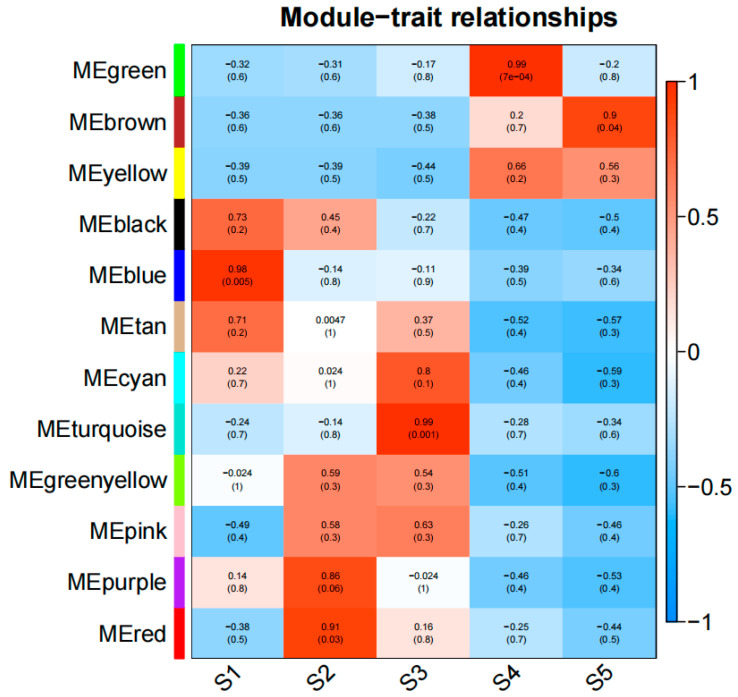
Heat map of correlation between sweet potato root development and metabolite expression patterns at different periods. The heat map shows the expression patterns of eight modules, with color bars representing the expression levels from high (red) to low (blue).

**Figure 10 genes-15-01319-f010:**
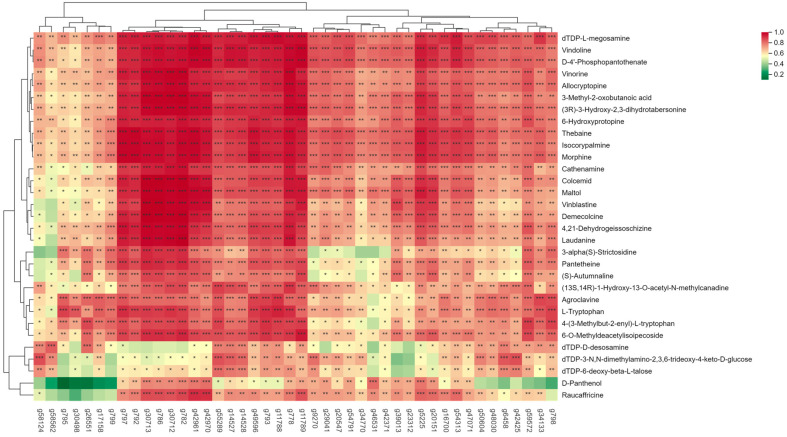
Gene-metabolite correlation analysis heat map. The color bar represents the level of correlation from high (red) to low (green). * represents the degree of correlation. * represents correlation, ** represents significant correlation, and *** represents extremely significant correlation.

**Table 1 genes-15-01319-t001:** Phenotypic indicators measured at five different developmental stages. S1, S2, S3, S4, and S5 represent the samples of days 7, 14, 21, 28, and 35 after planting, respectively.

Stage	Replica	Root Length (cm)	M ± SD	Diameter of Main Root (mm)	M ± SD	Number of Roots	M ± SD	Stem Length (cm)	M ± SD	Stem Diameter (mm)	M ± SD	Blade Weight (g)	M ± SD	Stem Weight (g)	M ± SD	Overground Weight (g)	M ± SD	Fresh Root Weight (g)	M ± SD
S1	1	23.8	21.9 ± 4.6	0.986	0.93 ± 0.22	9	11.8 ± 4.66	18.5	15.2 ± 3.78	4.38	4.58 ± 0.23	13.63	15.08 ± 2.11	2.91	2.5 ± 0.65	16.59	17.58 ± 1.68	4.86	6.28 ± 1.83
	2	26.2		1.276		14		15		4.33		16.22		2.48		18.69		8.96	
	3	18.3		0.856		8		12		4.84		18.02		1.97		19.98		5.27	
	4	25.4		0.676		9		19.5		4.76		12.71		3.36		16.06		4.91	
	5	15.8		0.836		19		11		4.61		14.81		1.78		16.58		7.41	
S2	1	42.1	37.86 ± 6.26	1.01	1.20 ± 0.11	13	15.6 ± 2.07	23	18.3 ± 6.14	4.36	4.59 ± 0.49	15.62	18.95 ± 2.7	3.68	3.5 ± 0.92	19.36	22.42 ± 3.15	12.18	14.53 ± 2.07
	2	40.8		1.22		14		26.5		4.97		21.39		4.83		26.26		16.52	
	3	34		1.254		16		15		4.37		19.15		3.46		22.65		14.08	
	4	28.8		1.304		17		15		5.21		21.73		3.29		24.66		16.82	
	5	43.6		1.19		18		12		4.03		16.88		2.26		19.18		13.07	
S3	1	41.8	34.51 ± 8.8	3.898	3.15 ± 0.61	12	10.8 ± 2.17	36	47 ± 15.86	5.3	5.93 ± 0.9	55.98	63.26 ± 27.18	14.15	19.38 ± 8.79	70.66	83.40 ± 36.27	36.25	32.45 ± 6.76
	2	45		2.364		13		40		6.67		43.74		14.6		59.07		28.14	
	3	25.14		2.844		12		51		5.27		71.91		20.18		92.5		40	
	4	33.7		3.608		9		73		7.14		106.1		34.37		142.31		34.75	
	5	26.9		3.03		8		35		5.28		38.56		13.6		52.48		23.09	
S4	1	27.6	26.62 ± 6.38	7.486	6.46 ± 1.03	16	10.8 ± 3.7	55	65.2 ± 21.46	6.38	6.13 ± 0.72	107.43	100.38 ± 31.4	23.09	32.69 ± 16.67	130.52	133.06 ± 46.39	48.17	37.64 ± 7.98
	2	27.5		7.204		13		94		6.4		136.04		48.74		184.78		42.59	
	3	16		4.846		7		48		5.91		63.68		12.54		76.22		28.1	
	4	33.2		6.238		8		47		5.03		72.3		28.07		100.37		32.38	
	5	28.8		6.52		10		82		6.94		122.43		50.99		173.42		36.95	
S5	1	19.4	21.28 ± 2.06	14.508	13.04 ± 1.44	15	16 ± 2.92	62	69 ± 9.43	7.59	6.54 ± 1.07	108.05	103.26 ± 28.81	34.98	34.49 ± 4.77	143.03	137.35 ± 33.75	88.07	88.86 ± 15.9
	2	23.5		12.922		16		77		7.81		151.5		42.44		193.94		116	
	3	22.9		12.308		12		58		5.64		82.3		33.25		113.55		78.22	
	4	21.7		14.39		17		68		5.86		84.36		31.17		115.53		85.37	
	5	18.9		11.088		20		80		5.78		90.07		30.61		120.68		76.66	

**Table 2 genes-15-01319-t002:** Quantitative indexes of internal structure of sweet potato differentiated roots on days 7, 14, and 21 of growth. The letters means the Significance.

Growing Days (d)	Transverse Section Diameter (μm)	Cross Section Area (μm^2^)	Cortical Thickness (μm)	Number of Primary Xylem	Number of Secondary Xylem
7	552.84 ± 29.44a	2.4 ± 0.26a	336.81 ± 44.77a	26.5 ± 4.2a	11.75 ± 2.75a
14	616.41 ± 88.93a	3.03 ± 0.9a	345.48 ± 48.21a	32.75 ± 3.77a	13.5 ± 3.7a
21	1063.96 ± 69.61b	8.91 ± 1.14b	614.49 ± 43.5b	81.5 ± 7.19b	20.75 ± 1.71b

**Table 3 genes-15-01319-t003:** Types and quantities of differential accumulation metabolites (DAMs).

Classification of Metabolites	Number
Carboxylic acids and derivatives	292
Prenol lipids	268
Fatty acyls	264
Organooxygen compounds	261
Steroids and steroid derivatives	111
Flavonoids	99
Benzene and substituted derivatives	93
Phenols	43
Indoles and derivatives	41
Cinnamic acids and derivatives	39
Organonitrogen compounds	35
Coumarins and derivatives	33
Hydroxy acids and derivatives	29
Purine nucleotides	28
Pteridines and derivatives	26
Purine nucleosides	24
Glycerophospholipids	23
Isoflavonoids	22
Pyrimidine nucleosides	22
Glycerolipids	17

## Data Availability

The original contributions presented in the study are included in the article/Appendix A, further inquiries can be directed to the corresponding author/s.

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
