# Peer review of "Integrated Transcriptional and Metabolomic Analysis of Factors Influencing Root Tuber Enlargement during Early Sweet Potato Development"

_genes, 2024, doi:10.3390/genes15101319_

Round 1

Reviewer 1 Report

Comments and Suggestions for Authors

Dear authors:

In the present study, the authors have combined transcriptome and metabolome analyses to investigate root/tuber enlargement in sweet potato Ipomoea batatas variety 'G8017’. The study findings help deeply understand the molecular mechanism of sweet potato root enlargement. The author has achieved satisfactory results through an appropriate analytical approach. However, the manuscript requires revisions to address several critical points and enhance the overall clarity of the study:

Title: Consider to rewrite it

“Combined transcription-metabolomics analysis of influence of  root tuber enlargement in early development of sweet potato”

Change to:

"Integrated transcriptional and metabolomic analysis reveals root tuber enlargement during the early development of sweet potato"

Or "Integrated transcriptional and metabolomic analysis of factors influencing root tuber enlargement during early sweet potato development"

Abstract:

This section is too long, so it should be shortened to about 200-300 words. Focus on concisely presenting the achieved results in the section Abstract.

Materials and Methods

- Section 2.3: A detailed description of the approach/method to performing the analyses for sucrose, soluble sugar and starch is required.

- Some access databases and tools/software need to be cited with the reference or access link. For example,  KOBAS data base (line 181).

Results

- Table 1: the phenotypic character of root tuber at five different developmental stages should be separated for each stage and include their average values ± SD. In addition, separate the “stage” collum into two columns (one for “stage” and one for “replica= the number of roots/plants, since S1-1 to S1-5 present the five different roots of each plant, then calculate the average values ± SD for each stage from five different roots.

- Fig1: lack the title/caption of y-axis. I recommended representing the Fig 1 data by dividing it into 3 catalogs of data sets such as:  Fig1a for the number of roots; Fig1b  for root size/dimension (root length,  stem length,  main root diameter, stem diameter ), and Fig 1c.  tuber weight  (remaining characters). 

- Fig2: the cross-sectional structure of differentiated root at the S5 stage should be included.

- It needs to be revised  (line 313-315): “there were 2411 (882 up-regulated, 1529 down-regulated) and 5214 (2770 up-regulated) genes in S1 vs S2, S2 vs S3, S3 vs S4 and S4 vs S5 groups, respectively.” I see here only the number of DEG for S1vs S2, and S2 vs S3 were presented. It lacks presenting the DEG number of S3 vs S4 and S4 vs S5.

- Table 3: should be removed since its data was presented in Fig4a.  There is an overlap in the data presented in Table 3 and Fig4a.

- No citation for data of Fig5 in Section 3.5.

- What does “positive ion mode “ and  “negative ion mode” mean? (line 374, 375)

- Table 4: clean up  the line number  (399, 400).

Some minor remarks:

- Since this phrase “differentially expressed genes” is a common term and used repeatedly in the article, its abbreviation should be used as (DGE).

- “mix 10 μ L of each sample into QC samples” (line 193-194): unclear meaning

- Rewrite:  “μ L” -> “μL”; 100C -> 100 oC; ect

- Using a uniform for ‘G8017’ or G8017. I think should be ‘G8017’.

- Italicize the gene name and the scientific name of the plant.

For example: Ibkn1, Ibkn2 and Ibkn3,..(line 59); CHS, ANS and 3GT (line 83) and the scientific name of the plant: Ipomoea batatas (line 10, 41), Arabidopsis  (line 515).

- Reformat the style according to the requirements of the journal, including:

+ Capitalize Each Word in the paper title and subsection following other journals’ style.

For example: “2.1. Experimental materials and treatment” -> “2.1. Experimental Materials and Treatment

+ Citation style of numbered ref: root crops[1]. ->  root crops [1].

I have marked some of the above comments on the manuscript. Please use it for easy tracking and revision.

Kind regards,

Comments on the Quality of English Language

Overall, the English in this manuscript is clear and easy to understand, except for a few sentences that are complex and unclear. I recommend a thorough review to address grammatical errors and sentence structure to improve the readability and clarity of the manuscript.

Author Response

Comments 1:[Title: Consider to rewrite it“Combined transcription-metabolomics analysis of influence of  root tuber enlargement in early development of sweet potato”Change to:"Integrated transcriptional and metabolomic analysis reveals root tuber enlargement during the early development of sweet potato"Or "Integrated transcriptional and metabolomic analysis of factors influencing root tuber enlargement during early sweet potato development".]

Response 1:[Thank you for pointing this out and I used the title "Integrated transcriptional and metabolomic analysis of factors influencing root tuber enlargement during early sweet potato development" as you suggested.Line2-4]

Comments 2:[Abstract:This section is too long, so it should be shortened to about 200-300 words. Focus on concisely presenting the achieved results in the section Abstract.]

Response 2:[Appreciate your suggestions.I whittled the summary down to about 250 words.Line14-34]

Comments 3:[Section 2.3: A detailed description of the approach/method to performing the analyses for sucrose, soluble sugar and starch is required.]

Response 3:[Thanks for the feedback.I have described the experimental method in detail in this paper.Line125-146]

Comments 4:[Some access databases and tools/software need to be cited with the reference or access link. For example,  KOBAS data base (line 181).]

Response 4:[Thank you for your comments.I have added the relevant references.Line196]

Comments 5:[Table 1: the phenotypic character of root tuber at five different developmental stages should be separated for each stage and include their average values ± SD. In addition, separate the “stage” collum into two columns (one for “stage” and one for “replica”= the number of roots/plants, since S1-1 to S1-5 present the five different roots of each plant, then calculate the average values ± SD for each stage from five different roots.]

Response 5:[Your comments are well taken. Revision done.Line269, Table 1.]

Comments 6:[Fig1: lack the title/caption of y-axis. I recommended representing the Fig 1 data by dividing it into 3 catalogs of data sets such as:  Fig1a for the number of roots; Fig1b  for root size/dimension (root length,  stem length,  main root diameter, stem diameter ), and Fig 1c.  tuber weight  (remaining characters).]

Response 6:[Thanks for your input. Changes are in the paper.Line270, Fig 1.]

Comments 7:[Fig2: the cross-sectional structure of differentiated root at the S5 stage should be included.]

Response 7:[Thank you for your advice. Due to the large diameter of the root during the S5 period, complete or 1/4 tissue could not be seen under the microscope, so anatomical analysis could not be carried out.]

Comments 8:[It needs to be revised  (line 313-315): “there were 2411 (882 up-regulated, 1529 down-regulated) and 5214 (2770 up-regulated) genes in S1 vs S2, S2 vs S3, S3 vs S4 and S4 vs S5 groups, respectively.” I see here only the number of DEG for S1vs S2, and S2 vs S3 were presented. It lacks presenting the DEG number of S3 vs S4 and S4 vs S5.]

Response 8:[Your opinion is important.  It has been modified. Line 331-334.]

Comments 9:[Table 3: should be removed since its data was presented in Fig4a.  There is an overlap in the data presented in Table 3 and Fig4a.]

Response 9:[Thanks for your comments. It has been deleted.]

Comments 10:[No citation for data of Fig5 in Section 3.5.]

Response 10:[Thank you for your guidance. It's been modified.Line 362]

Comments 11:[What does “positive ion mode “ and  “negative ion mode” mean? (line 374, 375)]

Response 11:[Thank you for your comment. It has been revised.Line 390.]

Comments 12:[Table 4: clean up  the line number  (399, 400).]

Response 12:[Thank you for your reminder. It has been deleted.]

Comments 13:[Since this phrase “differentially expressed genes” is a common term and used repeatedly in the article, its abbreviation should be used as (DGE).]

Response 13:[Your opinion is important.It has been modified.]

Comments 14:[“mix 10 μ L of each sample into QC samples” (line 193-194): unclear meaning.]

Response 14:[Thank you for pointing out. It's been modified.Line 208-209.]

Comments 15:[Rewrite:  “μ L” -> “μL”; 100C -> 100 oC; ect]

Response 15:[Appreciate your suggestions.It's all been modified.]

Comments 16:[Using a uniform for ‘G8017’ or G8017. I think should be ‘G8017’.]

Response 16:[Your advice is valuable. Changes implemented.]

Comments 17:[Italicize the gene name and the scientific name of the plant.For example: Ibkn1, Ibkn2 and Ibkn3,..(line 59); CHS, ANS and 3GT (line 83) … and the scientific name of the plant: Ipomoea batatas (line 10, 41), Arabidopsis  (line 515).]

Response 17:[Appreciate your suggestions. Revision made accordingly.]

Comments 18:[Reformat the style according to the requirements of the journal, including:+ Capitalize Each Word in the paper title and subsection following other journals’ style. For example: “2.1. Experimental materials and treatment” -> “2.1. Experimental Materials and Treatment”+ Citation style of numbered ref: root crops[1]. ->  root crops [1].]

Response 18:[I take your comments seriously and have made changes.]

Reviewer 2 Report

Comments and Suggestions for Authors

Congratulations on the impressive work and article!

The authors conducted a comprehensive study on sweet potato (Ipomoea batatas), focusing on the regulatory network of root enlargement through transcriptomic and metabolomic analysis during different early stages of root development. This research was complemented by phenotypic and anatomical observations.

Using RNA-seq technology, the authors identified 13,669 differentially expressed genes across five developmental stages of sweet potato roots. Notably, genes in the S1 vs S2, S3 vs S4, and S4 vs S5 comparison groups were enriched in the phenylpropanoid biosynthesis pathway. In addition, 67 differentially expressed transcription factors were identified, including those from the AP2, NAC, bHLH, MYB, and C2H2 families.

The metabolomic analysis detected a total of 4,695 metabolites, and K-means clustering revealed that lipids, organic acids, organic oxides, and other substances accumulated differently across the developmental stages.

Integrated transcriptomic, metabolomic, and phenotypic data indicate that the S3-S4 stage is crucial for sweet potato root development. Weighted gene co-expression network analysis (WGCNA) further classified transcriptome and metabolome data into distinct modules. KEGG enrichment analysis of highly correlated modules during the S3-S4 period showed that transcriptome differential genes were primarily associated with pathways such as fructose and mannose metabolism, pentose phosphate, caffeine metabolism, and others. The metabolomic data indicated a focus on amino sugar and nucleotide sugar metabolism, flavonoid and alkaloid biosynthesis, and more.

Based on WGCNA, 44 differential genes and 31 differential metabolites with high correlation were identified, shedding light on key changes in gene expression and metabolic activity during early root development, highlighting potential regulatory networks.

Suggestions:

Introduction: Limit to a maximum of four paragraphs.

Aim (lines 93-102): Make it more concise.

Chapter 2.3: Provide more details on sucrose, total soluble sugar, and starch content.

Figure 3: Clarify the significance (****).

Figure 6: Include more details about the statistical methods.

Figure 8: The right side (c, d, and e) is not clearly visible.

Figure 10: Specify the correlation symbols (*, **, ***).

Discussion: Make it more focused and concise.

This is a highly informative and well-structured paper!

Author Response

Comments 1:[Introduction: Limit to a maximum of four paragraphs.]

Response 1:[Thank you for your advice. The introduction is modified accordingly.]

Comments 2:[Aim (lines 93-102): Make it more concise.]

Response 2:[Your advice is valuable. Changes implemented.Line 91-97.]

Comments 3:[Chapter 2.3: Provide more details on sucrose, total soluble sugar, and starch content.]

Response 3:[Thank you for pointing it out. I have added the details of the experiment.Line 125-146.]

Comments 4:[Figure 3: Clarify the significance (****).]

Response 4:[Your opinion is important. Revision submitted.Line 315, Fig 3.]

Comments 5:[Figure 6: Include more details about the statistical methods.]

Response 5:[Thank you for your advice. The data in figure 6 is counted using the filtering and counting tools in word.]

Comments 6:[Figure 8: The right side (c, d, and e) is not clearly visible.]

Response 6:[Thank you for your feedback. However, there are a large number of related metabolites, please understand. The heat map in the attachment is enlarged more clearly.]

Comments 7:[Figure 10: Specify the correlation symbols (*, **, ***).]

Response 7:[Your opinion is important. Revision submitted.Line 479-480.]

Comments 8:[Discussion: Make it more focused and concise.]

Response 8:[Thank you for pointing out. It has been modified as much as possible.]